# Subjective Quality Assessment for Cloud Gaming

Abdul Wahab [1,*] , Nafi Ahmad [1] , Maria G. Martini [2] and John Schormans [1]

[1] School of Electronic Engineering and Computer Science, Queen Mary University of London, Mile End Road, London E1 4NS, UK; Nafi.Ahmad@qmul.ac.uk (N.A.); john.schormans@eecs.qmul.ac.uk (J.S.)

[2] School of Computer Science and Mathematics, Kingston University, River House, 53-57 High Street, Kingston upon Thames, Surrey KT1 1LQ, UK; mgmartini@ieee.org

* Correspondence: a.wahab@qmul.ac.uk

**Abstract:** Using subjective testing, we study the effect of the network parameters, delay and packet loss ratio, on the QoE of cloud gaming. We studied three different games, selected based on genre, popularity, content complexity and pace, and tested them in a controlled network environment, using a novel emulator to create realistic lognormal delay distributions instead of relying on a static mean delay, as used previously; we also used Parsec as a good representative of the state of the art. We captured user ratings on an ordinal Absolute Category Rating scale for three quality dimensions: Video QoE, Game-Playability QoE, and Overall QoE. We show that Mean Opinion Scores (MOS) for the game with the highest levels of content complexity and pace are most severely affected by network impairments. We also show that the QoE of interactive cloud applications rely more on the game playability than the video quality of the game. Unlike earlier studies, the differences in MOS are validated using the distributions of the underlying dimensions. A Wilcoxon Signed-Rank test showed that the distributions of Video QoE and Game Playability QoE are not significantly different.

**Keywords:** quality of experience (QoE); quality of service (QoS); packet loss ratios (PLR); cloud gaming; interactive gaming; video QoE; immersion; gaming influence factors

## 1. Introduction

Over the last couple of decades, rapid growth in communication technology has brought forward new services such as video streaming and cloud gaming over the internet. Due to the increased customer interest, tech giants such as Microsoft and Google have introduced their own cloud gaming platforms [1]. These platforms have millions of paid subscribers and the numbers of subscribers are increasing very rapidly.

This is a new form of gaming and is sometimes referred to as Gaming on Demand. The idea of cloud gaming is to play on a remote server and then stream the games over the internet to the user's device. The user's device does not need the hardware capability to run the game, since all the rendering of the game will be done at the server-side, saving the user the cost of getting a high-end GPU powered device [2]. This enables many devices such as phones, tablets, and PCs to run high-end hardware intensive games at satisfactory quality.

Real-time applications like cloud gaming are quite demanding in terms of Quality of Service (QoS) parameters, requiring high bandwidth and low latency and packet loss ratio (PLR) conditions [3]. Service providers are aware of the potential growth of new services and are keen to keep their existing customers happy as well as gain new customers. In order to do that, they have to provide competitive service level agreements to ensure satisfactory Quality of Experience (QoE) for cloud gaming applications.

QoE assessment of cloud gaming applications is a trending research field. With very limited literature on the subject, the field and the tools employed for QoE evaluation of cloud gaming applications are far from maturity. There are accounts of subjective testing being used to evaluate the effect of cloud gaming influencing factors, such as video encoder, frame rate and QoS parameters [4]. However, as game development is becoming more

sophisticated, it is vital to keep up the QoE evaluation targeted for more complex and high-end games.

Like any other emerging application, the QoE of cloud gaming can also be evaluated subjectively and objectively. The subjective quality assessment includes the employment of human subjects that rank the quality of their perception of the game. This is a time-intensive and resource-intensive task. Alternatively, the objective evaluation uses application-based key performance indicators (KPIs) to evaluate the quality of perception. For instance, in the QoE assessment of video streaming applications, this technique employs algorithms such as PSNR, SSIM and VMAF to evaluate the QoE of the service, see [5,6]. These objective metrics are derived or correlated to the subjective QoE to assess the QoE of any application accurately and robustly. Since cloud gaming is a relatively new application, the objective metrics are not completely mature and hence mainly rely on subjective analysis.

In this paper, we used subjective testing to evaluate the effect of various network conditions employing a novel emulator on the video quality, game playability and overall experience of the user for three games with different characteristics in terms of genre, content complexity and pace. The results provide indications on which factor (video quality or the game-playability ) contribute more towards the overall QoE of the games using MOS and distributions of each QoE dimension. We also demonstrated the shortcomings of using traditional Mean Opinion Score (MOS) using statistical tests on QoE distributions.

Our results add significant value to existing subjective QoE studies by proving the difference between video quality and game playability quality using distributions and statistical tests. For instance, in [4,7], the authors used MOS to evaluate the effect of network parameters on QoE of cloud gaming. These studies used MOS as a measure of QoE and presented the differences between game QoE and video QoE of cloud gaming applications. Their analysis does not address the shortcomings of MOS that tend to ignore the user diversity of the underlying sample. Moreover, the literature report MOS alone is not adequate to assess QoE accurately [8]. These shortcomings are addressed in this paper, and a novel analysis of video QoE and game QoE was presented in the light of existing studies.

The remaining paper is organised as follows: In Section 2, we present related works accounting for the existing studies on the QoE evaluation of cloud gaming. Section 3 presents the details of the Games under Test (GuTs) and the selection criteria for selecting these games. Section 4 highlights the methodology undertaken for this paper. This includes the details of the testbed and the subjective study carried out. In Section 5, we present the results of the experiments for all the network scenarios tested. The results are discussed in Section 6. Finally, Section 7 concludes this paper and presents a brief account of future work.

## 2. Related Work

Improvement in the cloud infrastructure and increased interest by the users have propelled academia and industry to address QoE evaluation of cloud gaming applications. Authors in [2] describe the infrastructure of cloud gaming. They discuss how powerful machines in the cloud can host the game and stream it to the client's machine over the network. The network transfers the video traffic from the cloud to the user and the user's input to the cloud. This makes the network QoS parameters an important influencing factor in QoE evaluation of such service. They also presented a review of different techniques (subjective and objective) to evaluate the QoE of cloud gaming.

The two main factors impacting the overall QoE of cloud gaming are the video quality and the quality of interactivity of the user with the game. The video quality focuses on the graphics and frame dynamics of the game, whereas the game quality assesses the interactive dynamics such as ease of movement and reactivity of the gameplay to the user command. Studies in [5,9] show that video QoE tends to degrade significantly with deteriorating network performance. They reported how increasing levels of PLR result in lower QoE for passive gaming applications. However, they did not study the effect of

QoS parameters (delay and PLR) on the interactive gaming applications, where the user interacts with the games and the traffic is exchanged over the network.

Claypool et al. [10] referred to the gaming client as a thin client and studied detailed network performance when running different games on the onLive platform. Their work was extended by the authors in [11] to study the effect of various network conditions on the objective quality metrics of two different cloud gaming platforms. They concluded that network degradation affects video and game quality on both platforms, but the onLive platform performs better than StreamMyGame (SMG) at degrading network conditions. They used frame rate as a measure of game quality. The authors tested traditional games employing low (24, 30) frames per second (fps) that are not representative of current games hosted on cloud gaming platforms. This paper studies a more recently deployed platform (Parsec) and evaluate modern games with a higher frame rate (60 fps).

In addition, the impact of the network parameters on the cloud gaming scenario was also reported in [7,12]. They replicated the cloud gaming scenario using the PlayStation 3 console as a cloud machine and studied the effect of different network scenarios on the QoE of the game. They reported that at PLR > 1%, the QoE perceived is unsatisfactory. Moreover, a similar study [4] demonstrated the effect of the network conditions (frame loss, jitter, and latency) hosted on NVIDIA GeForce Now. They used subjective testing to evaluate QoE and reported that the games perform the worst for increasing delay but show resilience to packet loss. Furthermore, authors in [13] showed how varying bit-rates and frame rate can affect the user perception of cloud gaming. They reported using MOS that increasing frame rate and bit-rate enhances user quality but require better network to support excessive traffic generated by increasing frame rates and bit rates.

Above mentioned studies used MOS to reach their conclusion about the effect of network QoS on video and game experience. However, other studies such as [8,14,15] demonstrated that MOS can be deceiving when analysing QoE of such applications since it does not reflect the diversity of user ratings. Moreover, they reported that, the MOS is not an ideal representative of an ordinal Absolute Category Rating (ACR) scale as used in subjective QoE studies. We presented our results as MOS and then validated them with an analysis of the underlying distributions. Moreover, we used a bespoke emulator that replicates network traffic realistically, unlike the traditional emulators that use static mean delays.

Furthermore, authors in [16] have presented some Key Quality Indicators (KQIs), such as frame rate and frame rendering time Round Trip Time (RTT), for objective QoE evaluation of cloud gaming. They studied the effect of the type of transport network on the presented KQIs. They reported that Ethernet is the best transport network to cloud gaming in comparison with Long Term Evolution (LTE) and WiFi networks. However, they did not study the effect of QoS parameters offered by these transport networks on the objective KQIs of cloud gaming.

In our paper, we use Parsec as a game hosting platform and study how the different network settings impact the video and game-playability of the user. As far as we know, there are no other studies available on Parsec. Additionally, instead of using traditional emulation with static mean delays, we use a realistic lognormal delay distribution to emulate delay. Moreover, we use MOS and underlying distributions to evaluate which factor (video or game-playability) is more critical in the overall QoE evaluation of cloud gaming.

## 3. Games under Test

Three different games were selected on the basis of genre, pace, content complexity and the popularity of the gaming platform. The selection was made in accordance with International Telecommunication Union-Telecommunication (ITU-T) recommendation G.1032 [17]. As there is very limited literature on the classification of the games [18], we tested various games and evaluated them on the basis of the aforementioned influencing factors. The series of steps to select the games and narrow them down to three games are shown in Figure 1.

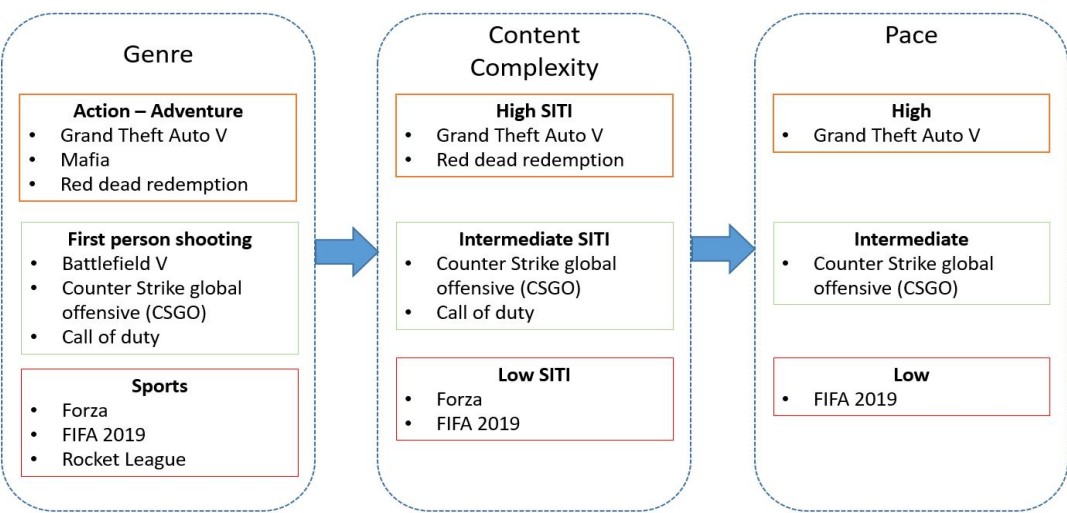

**Figure 1.** Stages of GuT selection.

The games with most distinctive difference in content complexity and pace were selected. This is because, according to [17], these influencing factors are critical when studying the effect of QoS parameters, e.g., delay and PLR. Figure 1 shows a set of influencing factors against which the games were selected. We tested three popular genres of games: Action-adventure, First-Person-Shooting (FPS), and Sports. In the first stage, we selected three most popular games on Twitch.tv of each genre. In the next step, we calculated content complexity (SI and TI) of five scenarios of each game and then filtered the outliers. Finally, the remaining games were tested for their pace and three games; one of each genre, was selected for subjective testing. These stages are summarised in Figure 1.

The selected games represent a broad range for each genre. As there are no set guidelines to define the pace of a game, we evaluated the pace of the game on the basis of response time required by the user and the temporal complexity (explained later). Additionally, the learning difficulty of the game can also be an influencing factor [17]. This was assessed on the basis of the number of different moves and behaviours a user needs to learn to play the game. Grand Theft Auto V (GTA V) has multiple modes, but, in this paper, we only considered the driving scenario, whereas, generally for FIFA and Counter Strike Global Offensive (CSGO), a standard scenario includes most of the aspects of the gameplay.

*Spatial and Temporal Information of GuT*

To evaluate the content complexity of the different games, we calculated spatial information (SI) and temporal Information (TI) of the recorded scenario of the game play using ITU-T recommendation P.910 [19]. We calculated SI and TI for five different scenarios at a resolution of $1920 \times 1080$ for each game consisting of the gameplay that would be evaluated by subjective testing. The SI is calculated for each frame on the basis of the standard deviation of the luminance component of that frame, whereas the TI is calculated between successive frames and hence can be used to represent the change in pixels with respect to time. Both SI and TI evaluate content complexity in terms of space and time, respectively. The SI and TI were calculated for all the frames, but the final results were the maximum SI and TI of the frames as suggested by [19]. Figure 2 shows the SI and TI of all the videos used in this paper.

The SI and TI of GTA V videos are higher than in CSGO and FIFA. This is because, in GTA V, the adjacent frames are very different and hence the TI was higher than the other games. This was one of the reasons we classified GTA V as a fast-paced game.

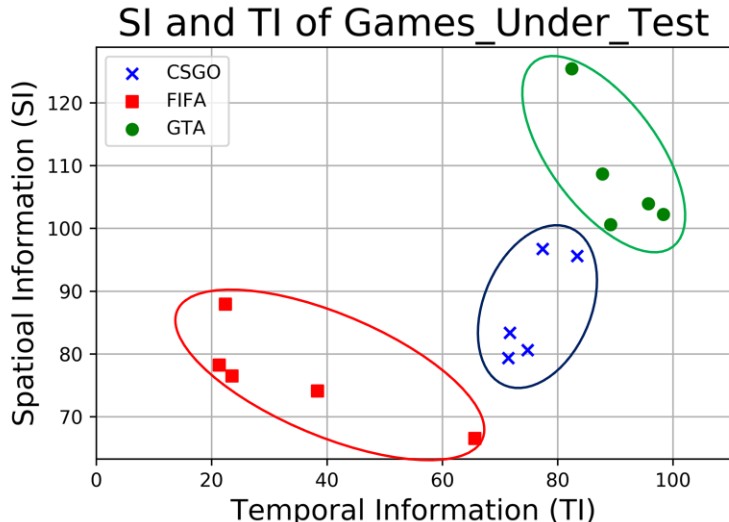

**Figure 2.** SI and TI of GuT.

The content complexity of FIFA was lowest, and this results in lower values of SI and TI for most videos tested. This is because most of the frame content is of the grass pitch observed from a broadcast camera angle. Thus, the content of the frames is simpler and results in a low SI. Additionally, only a very small proportion of the frame changes with time, resulting in the lowest TI among other GuT. On the other hand, the content complexity of CSGO is between FIFA and GTA V. This is due to the first-person perspective that can result in faster changes within adjacent frames giving it a higher TI than FIFA. Moreover, since the user is walking/running and not driving, the changes are slower than GTA V and hence this game has a lower TI than GTA V.

## 4. QoE Evaluation Using Subjective Testing

Subjective testing was carried out to determine the QoE of three GuT at different network conditions. In this paper, we used three different types of network scenarios that are: (1) Delay-Only, (2) PLR-Only, and (3) Mixed (Delay + PLR). We also captured three dimensions of the quality that are: Video-Quality (V-QoE), GamePlayability-Quality (GP-QoE), and Overall-Quality (O-QoE). All of these dimensions were independently captured. Figure 3 shows the summary of the methodology undertaken in this paper and also shows the processes that take place in cloud and client side of the setup.

These three categories of network scenarios were selected because they represent network conditions experienced in real networks. In real networks, delay is experienced by all traffic, but this can be without any loss of packets, thus showing delay-only scenarios. Moreover, there are also instances in the network when packet loss occurs at very low/insignificant delay as seen in 5G networks and thus justifies the selection of PLR-Only scenarios. Furthermore, both of these QoS parameters also exist together in the network, see [9]. All of these QoS parameters were implemented on the ingress interface of the client coming from the cloud, thus mimicking the downlink traffic of the client of a cloud gaming application.

The delay used in this paper is not the traditional mean static delay as used in previous work [4,7]. We used lognormal delay distributions to emulate more realistic network scenarios. Our previous work [9] shows that QoE of video applications yields different results for static mean delay and delay distribution, hence delay distribution was implemented in this paper to replicate a more realistic cloud gaming scenario. The parameters of the lognormal distribution were calculated using targeted mean delay and a standard deviation of <1 ms. This was to ensure that the effect of jitter is minimal [12,20,21]. The details are not the focus of this research and can be seen in [9]. Linux based emulator NetEm was used to configure different network scenarios, as shown in Table 1.

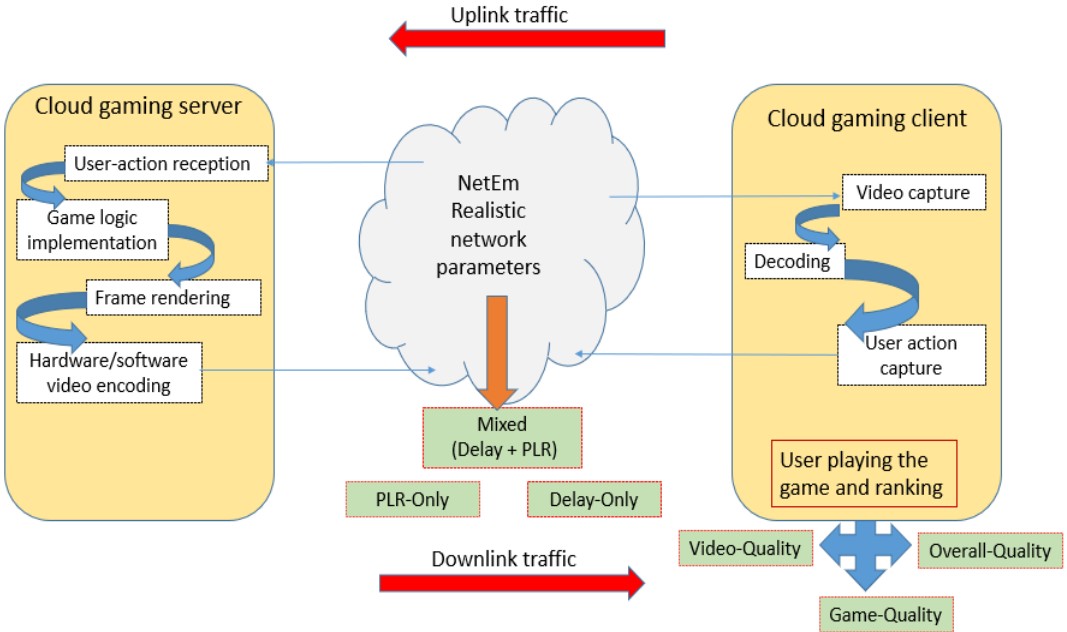

**Figure 3.** Block diagram of processes taking place on the client and server side and the flow of traffic.

**Table 1.** Summary of network scenarios.

| Network Scenario | Magnitude | |
|---|---|---|
| Delay-Only (ms) | 10, 20, 30, 50, 75, 100, 200, 300 | |
| Packet Loss Ratio only (%) | 0.10, 0.25, 0.50, 1, 2, 5, 10, 20, 25 | |
| Mixed (Delay (ms) \| Loss (%)) | 10 | 0.10, 0.25, 0.50, 1, 2, 5, 10 |
| | 30 | 0.10, 0.25, 0.50, 1, 2, 5, 10 |
| | 50 | 0.10, 0.25, 0.50, 1, 2, 5, 10 |

The magnitudes of the QoS parameters were selected after surveying similar studies [2,4]. In these studies and our preliminary findings, we observe that, at these QoS parameters, the QoE of the gaming applications shows variation. Therefore, these were the key parameters at which the QoE was evaluated.

The cloud gaming scenario was replicated using a three PC setup, see Figure 4. The machine designated as cloud (M1) has higher specifications (Intel i9 processor, RTX 2080ti GPU) in comparison with the client machine (Intel i5 processor, integrated GPU) (M2), where the game was played. Both of these machines were connected via a third machine running NetEm. The traffic coming from M1 was bridged to NetEm, taking the traffic to M2 and vice versa. NetEm was used to configure different QoS parameters, thus mimicking the internet connection between the cloud and client machine as experienced in the real-life scenario.

On both machines, Parsec was used to run the game. Parsec is a Desktop capturing application that is widely used for remote gaming applications. It provides better infrastructure for gaming applications. One reason for that is that Parsec transports network traffic over Better-UDP (BUD), as the transport layer protocol developed by Parsec makes it more suitable for game network traffic. This provides advantages over other remote-play platforms [22].

Parsec was setup on both PCs and changes were made to the configuration file to ensure that the traffic was routed via the Ethernet to the NetEm box. Table 2 shows game parameters that were used in this study.

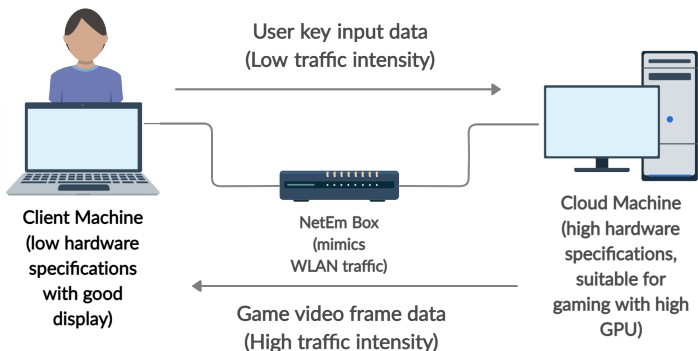

**Figure 4.** Cloud gaming test-bed.

**Table 2.** Summary of Parsec streaming parameters.

| Resolution | $1920 \times 1080$ |
|---|---|
| Bandwidth | 10 Mbps |
| Encoder | H.264 |
| Frame rate | 60 fps |

*Subjective Testing and User Profile*

In this paper, subjective testing was performed to measure QoE. 22 subjects were employed. Subjects were tested for visual acuity using a Snellen chart and none of the subjects reported any visual impairments. The mean age of the subjects was 27.3 years, median age of 28, and standard deviation of age of 4.5. All the subjects had a general knowledge of operating computers and were familiar with gaming. There were four females and 18 males in this investigation. These experiments were carried out in November when the lockdown rules were not very strict in the UK.

Before the start of the experiment, the subjects were told about the general mechanics of the game. These include the rules, general gameplay and control instructions. In addition to that, a short time was provided to get used to the game. The subjects played a game on M2 that had an ASUS 24″ ($1920 \times 1080$ p) FHD display. The distance between the user and the screen was maintained according to [17], and the lighting of the room was regulated according to ITU-T recommendation BT.500-13 [23]. Each subject was asked to play each setting of the network scenario per game for at least 60–90 s in accordance with ITU-T recommendation P.809 [24]. The network conditions that were tested in the first five tests were repeated again during the test to ensure that the game practice does not affect the rating. This is because, at the beginning of the test, the users are less familiar with the game dynamics and might rank differently. To minimise the effect of this, the mean of the two results was taken as a measure of QoE in those network conditions. Each subject tested 38 different network conditions, see Table 1, per game that took them around 60–90 min to complete. One game was tested in one sitting to avoid subject fatigue.

After completing each scenario, the subject was asked to rate the V-QoE, GP-QoE and O-QoE on a 7-point ACR scale as recommended in [24], see Figure 5. The V-QoE demonstrates the graphics of the game, the video quality perceived by the user, and can be judged passively. However, GP-QoE indicates the game response to the user inputs, ease at which game can be played (this is not the user's ability to play the game) and players' interactivity with the game. The subjects were briefed to assess both of these QoE dimensions independently from each other. Finally, O-QoE is the overall perception of the game for that specific scenario. O-QoE is not the mean of V-QoE and GP-QoE.

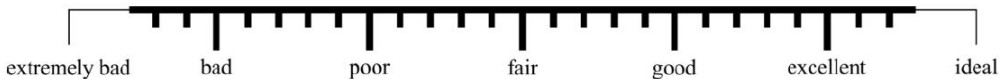

**Figure 5.** 7-point Absolute Category Rating (ACR) scale [24].

## 5. Results

The results obtained by subjective testing were used to calculate the MOS of each game under different network scenarios. We calculated MOS of V-QoE, GP-QoE and O-QoE for each network scenario. Moreover, we present our results in three classes of network scenarios that are: (1) Delay-Only, (2) PLR-Only and (3) Mixed Scenario (Delay + PLR).

### 5.1. Delay-Only Scenario

As expected, the QoE of all GuT shows a downward trend for increasing delay. The consistency in the trend is seen for all different dimensions of quality, see Figure 6. However, the QoE of all three GuT is affected differently for increasing magnitudes of delay. For instance, FIFA performed significantly better (up to 53%) than the other two GuT for all QoE dimensions, whereas it is evident from Figure 6 that GTA performed the worst for V-QoE. Moreover, CSGO and GTA have similar GP-QoE and O-QoE for increasing delay.

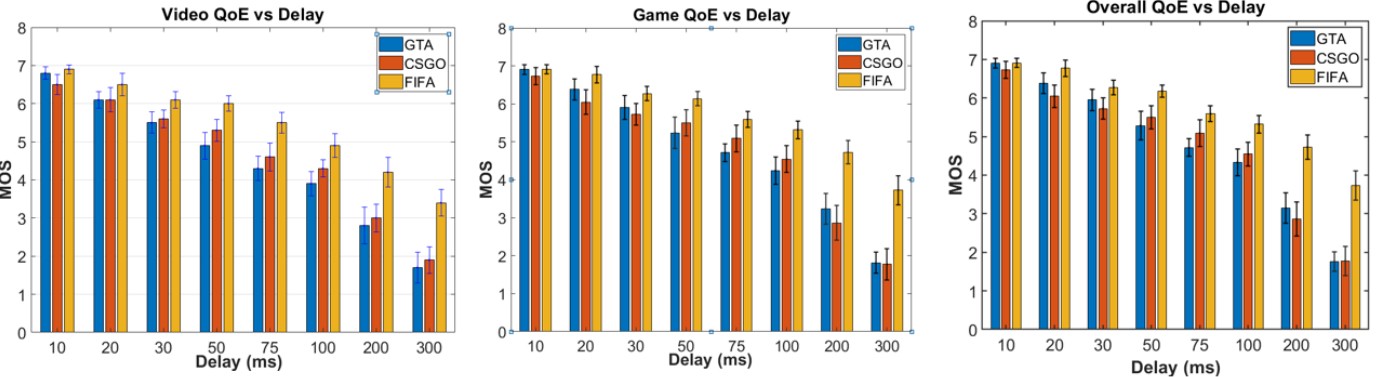

**Figure 6.** Different QoE dimensions against delay-only of GuT.

This can be explained using the content complexity and the pace of the game. FIFA has the lowest SI and TI and is relatively slow-paced among all GuT, see Figure 2. This means that, compared to the other games, the consecutive frames in FIFA have higher similarity (lower TI). Hence, even at higher delay, V-QoE of FIFA is less affected than other GuT. Moreover, GTA has higher content complexity (SI and TI) resulting in a lower V-QoE compared to other GuTs for increasing delay.

Additionally, we found that GP-QoE depends on the pace of the game, level of user interactivity, and the gameplay of the game. For instance, in FIFA, the user spends a considerable amount of time in slow paced actions such as passing and dribbling. This means that the user has more time to register moves, and thus delay has a lower impact on game-playability of slowed paced games such as FIFA, whereas CSGO and GTA have a faster gameplay and require quicker action such as aiming and driving. This means the user needs to interact/react quicker to changing scenarios of the game. In this case, a small delay at a crucial moment of the game can result in reduced GP-QoE.

Another interesting observation was that the O-QoE follows GP-QoE closely for delay scenarios for all GuT. This can be explained with respect to the level of interactivity and immersion in the game that is experienced by the user. For instance, in decisive moments of the gameplay, subjects showed an inclination towards game-playability rather than the video quality. Consequently, when the video quality is bad, but the user is still able to play fine, the user tends to rate the overall quality higher. The opposite of this user scoring behaviour was rarely seen in this investigation.

This holds true when the MOS of different QoE dimensions were compared. When confidence intervals were investigated, an overlap between different quality dimensions was observed. Since QoE is ranked on an ordinal scale, MOS alone will not be a conclusive metric to find the significant difference between V-QoE and G-QoE. This requires further statistical evaluation, which is discussed in the later Section 6.

### 5.2. PLR-Only Scenario

Similar to the delay-only scenarios, the QoE values of all GuT show a downward trend for increasing PLR. The consistency in the trend is seen for all different types of QoE, see Figure 7. The QoE of all three GuT is affected very similarly by the increasing levels of PLR. However, GTA still performed the worst out of the three GuT for all types of QOE and at PLR > 1%.

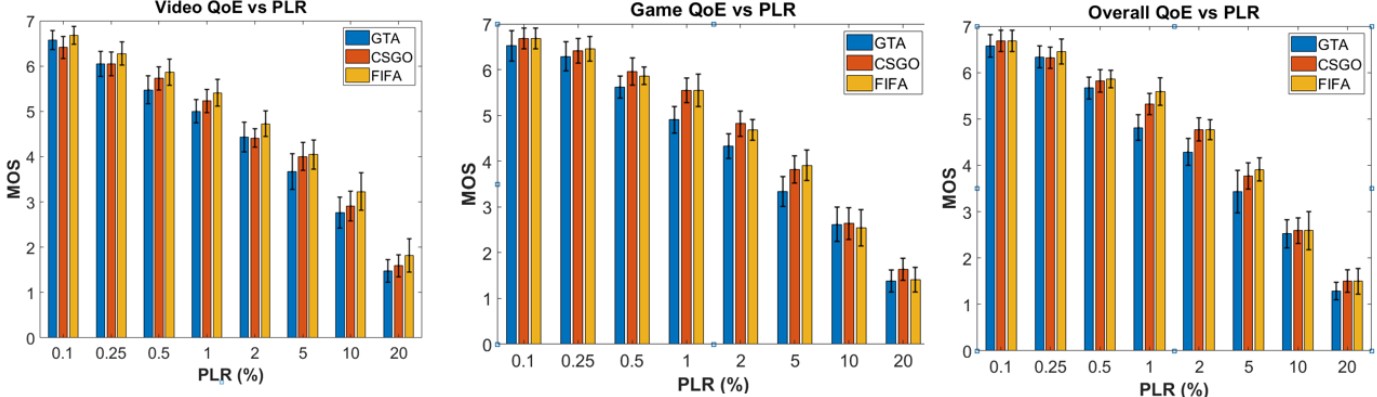

**Figure 7.** Different QoE dimensions against PLR-only of GuT.

An interesting finding was about the V-QoE of all games for increasing PLR. The results show more a significant difference than the results reported in literature. As reported in [4], for PLR > 1%, the V-QoE of the FPS shooting game (like CSGO) degraded heavily. In contrast, all games including CSGO showed a lot of resilience to PLR > 1% up to 10%. This can be explained using error corrective mechanism employed by the games/platforms. The improved error concealment methods such as frame interpolation can reconstruct part of the missing frames with significant improvement [4,22]. This means that, even when a specific part of the frame is lost, it is reconstructed by using other available parts of the frame.

The GP-QoE indicates a similar trend to V-QoE for all games, but the GP-QoE did not reach MOS = 1, even at PLR = 25%. GTA V game-playability degraded the most and a high proportion of subjects reported occasional loss of control in scenarios with PLR > 5%. For FIFA, the GP-QoE decreases more in comparison with the V-QoE for increasing PLR. This is due to the jerkiness and blockiness introduced by the PLR to the video quality that makes it difficult for the users to control and coordinate the ball between the players. We found that for PLR > 5% the GP-QoE was highly affected due to video artefacts. In [5], it is reported that FIFA's video quality degraded sharply for increasing PLR levels, whereas, for CSGO, the GP-QoE shows a very similar result to V-QoE.

Another interesting finding was that O-QoE of all the games did not follow any specific quality factors (V-QoE or GP-QoE). It can be seen in Figure 7 that, for PLR < 10%, the O-QoE follows the G-QoE closely as the user perception of the overall quality was more affected by the ease of the play and the interactivity of the game, whereas, for PLR > 10%, the O-QoE follows the V-QoE more closely. This shows that, when the V-QoE was bad, but the user was still able to interact with the game, users tend to rate the O-QoE lower than the subsequent GP-QoE. This analysis is valid when comparing the quality using MOS. As reported earlier and in [14], MOS has limitations when comparing two metrics

measured using the ordinal scale. Hence, the distributions of each dimension need to be compared to make a valid conclusion about their differences.

*5.3. Mixed (Delay + PLR) Scenario*

We report in this section the results of the evaluation of the three types of QoE for mixed scenarios i.e., combination of delay and loss scenario. Seven levels of PLR conditions were evaluated for each 10 ms, 30 ms and 50 ms delay, as reported in Table 1. The results are presented as heatmaps (see Figures 5–7), where the colour of the block shows MOS at that network parameter combination. A darker colour shows an MOS closer to 1 (extremely bad quality) and a lighter colour shows an MOS closer to 7 (ideal quality).

To elaborate the findings, we also present V-QoE (MOS) as Good-to-Better (%G2B) and Poor-to-Worse (%P2W) ratios in Table 3, where %G2B is the number of network conditions at which the MOS > 4 out of all mixed network scenarios. The same is true for %P2W for MOS < 3. The higher %G2B and lower %P2W indicate better quality and vice versa.

**Table 3.** Summary of %G2B and %P2W of different QoE dimensions for GuT.

| QoE Dimension | GuT | %G2B | %P2W |
|:---:|:---:|:---:|:---:|
| **Video Quality** | GTA | 48 | 33 |
| | CSGO | 57 | 24 |
| | FIFA | 67 | 19 |
| **Game Playability** | GTA | 52 | 29 |
| | CSGO | 57 | 19 |
| | FIFA | 62 | 24 |
| **Overall** | GTA | 57 | 33 |
| | CSGO | 57 | 19 |
| | FIFA | 62 | 24 |

5.3.1. Video QoE

As expected, for all three games, the V-QoE decreases with increasing delay and loss conditions, see Figure 8. Even in mixed conditions, GTA performed worst among all GuT for every mixed scenario. Similarly, the V-QoE of FIFA degraded least with CSGO in between the other two GuT. This clearly indicates how the content complexity plays a vital role in V-QoE of cloud gaming applications. V-QoE of games with higher content complexity are affected most by worsening QoS conditions.

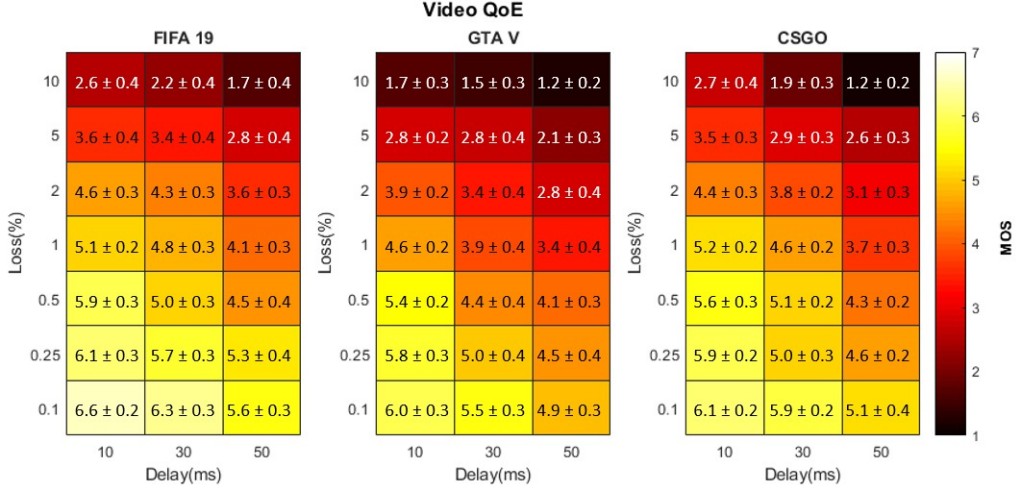

**Figure 8.** Heatmap of V-QoE (MOS with Confidence Intervals) for mixed network scenarios.

It is observed that GTA V has low %G2B for V-QoE as compared to CSGO and FIFA. It also has the highest %P2W among the three games. This confirms that V-QoE of GTA was most affected by the combination of delay and PLR. FIFA has the highest %G2B and lowest %P2W, showing that V-QoE of FIFA was least affected by the network impairments among the other games. In addition to this, CSGO V-QoE is intermediate between GTA V and FIFA. Similar findings are reported in [7], where the authors concluded that fast-paced games perform worse for increasing delays as compared to slower games. On the other hand, the slow games get affected more by the PLR than increasing delay.

### 5.3.2. Game QoE

Once again, GTA V has the lowest GP-QoE out of all three games for all network scenarios, see Figure 9. This can be explained due to the pace of the game and hence worsening network conditions affect the user response more significantly. These results are similar to the results presented in [12]. This indicates that the MOS of fast paced games degrades more significantly for worsening network conditions than slow-paced or medium paced games regardless of the frame rate, resolution and gaming platform.

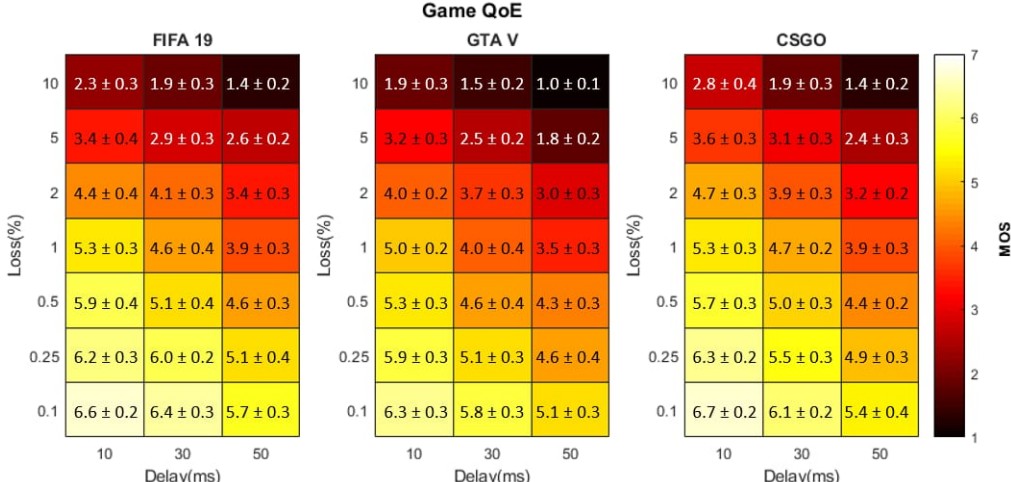

**Figure 9.** Heatmap of GP-QoE (MOS with Confidence Intervals) for mixed network scenarios.

Unlike V-QoE, FIFA showed lower GP-QoE at higher levels of delay and PLR in comparison with CSGO. This is because of the slow-pace of the game. As reported earlier, slow paced games degrade more for increasing PLR levels. This means that, at high packet loss, the user interaction with the game gets affected. Consequently, the user tends to rank a lower quality of perception.

This is seen in the second row of Table 3, where %P2W of FIFA is 24% and CSGO shows improvement with a %P2W of 19%. In contrast, at lower levels of delay and PLR, FIFA has a better GP-QoE than CSGO. This is observed in Table 3, where FIFA has %G2B of 62% as compared to 57% for CSGO.

These findings are only valid, when comparing the MOS of two games. As reported earlier and in [8,14], MOS has inherent limitations and is not an ideal metric to compare QoE. To address these limitations and draw better conclusions about the differences between the V-QoE and GP-QoE, it is essential to compare the underlying distributions of these QoE dimensions.

### 5.3.3. Overall QoE

Finally, the O-QoE of the GuT was evaluated. All games showed a degradation in O-QoE for increasing levels of QoS parameters, see Figure 10. GTA V showed the lowest at all network scenarios while CSGO and FIFA showed conflicting results at lower and higher magnitudes of PLR as seen in GP-QoE. We used %G2B and %P2W of all three QoE dimensions to evaluate which one out of V-QoE or GP-QoE more closely correlates to the O-QoE.

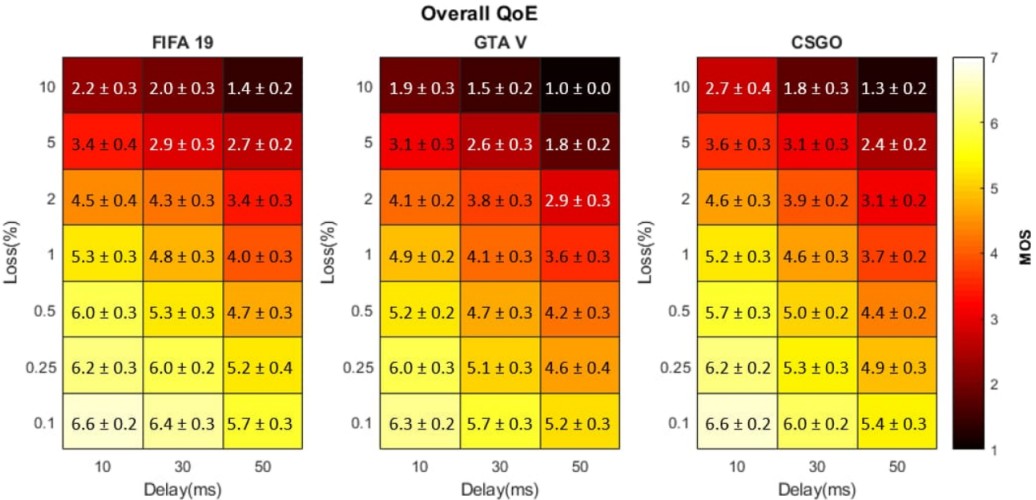

**Figure 10.** Heatmap of O-QoE (MOS with Confidence Intervals) for mixed network scenarios.

It is observed that, for both %P2W and %G2B, O-QoE shows a similar trend as seen in the GP-QoE for all games. This confirms that game playability is a stronger influencing factor than video quality of overall quality perceived by the user. These findings are critical for QoE evaluation of cloud gaming applications and need to be evaluated further. Even though the MOS of two dimensions (V-QoE and GP-QoE) relates differently to O-QoE, it is important to compare other QoE metrics to draw a valid conclusion. As reported in [8], the mean is not an ideal statistical measure when comparing ordinal quantities. Therefore, the underlying distributions of V-QoE and GP-QoE were compared using statistical tests.

## 6. Discussion

To evaluate the difference between V-QoE and GP-QoE, it is important to consider the underlying distributions as MOS alone is not representative of the whole sample size. When the MOS were compared, V-QoE and GP-QoE at most network conditions show differences. However, when the confidence intervals were compared, an overlap was seen between the instances of both V-QoE and GP-QoE. Therefore, we extended our analysis and carried out Wilcoxon Signed-Rank (WSR) tests to evaluate the difference between the two aforementioned QoE dimensions. Studies such as [4,7,12] have reported differences between V-QoE and GP-QoE using MOS alone and never analysed the underlying distributions to validate their findings. This is the first such study where we attempted to validate the differences in MOS using distributions of the underlying QoE dimensions.

Since QoE is measured using an ordinal ACR scale, the best candidate for the statistical test to compare these QoE dimensions is the WSR test. This is because it is valid to compare ordinal quantities when the two samples are related. In our case, since the same subjects ranked V-QoE and GP-QoE, the samples were dependent. This does not mean that V-QoE and GP-QoE have any overlapping features. They were evaluated on separate criteria and have no overlap between them.

The two samples: V-QoE and GP-QoE are the response variables of the investigation, while the ACR scale is the explanatory variable. The NULL hypothesis of the investigation was that V-QoE and GP-QoE have the same distribution and have no significant difference between them. The WSR test was carried out on both of these samples for each game in all the network scenarios. The results of the test were presented in Figure 11. We used a critical value of 5% as it is used widely in the studies employing the WSR test. If the WSR value at a specific network scenario is greater than the critical value, the NULL hypothesis was accepted, meaning that V-QoE and GP-QoE have the same distributions and any differences are due to chance. Alternatively, if the WSR value is lower than the critical value, the NULL hypothesis is rejected and both dimensions (V-QoE and GP-QoE) have significant differences. In Figure 11, the green boxes indicate the network conditions at

which the NULL hypothesis is rejected, while the red boxes show the network condition in which there are no significant differences between the V-QoE and GP-QoE.

| | Game\delay | 10ms | 20ms | 30ms | 50ms | 75ms | 100ms | 200ms | 300ms |
|---|---|---|---|---|---|---|---|---|---|
| **Delay Only** | GTA | | | | | | | | |
| | FIFA | | | | | | | | |
| | CSGO | | | | | | | | |

| | Game\PLR | 0.1% | 0.25% | 0.5% | 1% | 2% | 5% | 10% | 20% | 30% |
|---|---|---|---|---|---|---|---|---|---|---|
| **PLR only** | GTA | | | | | | | | | |
| | FIFA | | | | | | | | | |
| | CSGO | | | | | | | | | |

| | 10ms Delay | | | | | | | |
|---|---|---|---|---|---|---|---|---|
| | Game\PLR | 0.1% | 0.25% | 0.5% | 1% | 2% | 5% | 10% |
| **Mixed** | GTA | | | | | | | |
| | FIFA | | | | | | | |
| | CSGO | | | | | | | |
| | 30ms Delay | | | | | | | |
| | GTA | | | | | | | |
| | FIFA | | | | | | | |
| | CSGO | | | | | | | |
| | 50ms Delay | | | | | | | |
| | GTA | | | | | | | |
| | FIFA | | | | | | | |
| | CSGO | | | | | | | |

**Figure 11.** Wilcoxon Signed-Rank tests results for each GuT at different QoS parameters. Red boxes show that the Null hypothesis is accepted, and Green boxes show that the Null hypothesis is rejected.

It is clear from Figure 11 that, in most of the network conditions, the results of WSR were greater than the critical values. This means that, for most of the network combinations, there was no significant difference between the V-QoE and GP-QoE for all GuTs. Alternatively, in a few network conditions, WSR value was smaller than the critical value and indicates that there were significant differences between the two QoE dimensions. The ratio of green to red was 12/121. This means that, for almost 91% of the times, the NULL hypothesis was accepted. This signifies that the differences between V-QoE and GP-QoE are not significant. This also means that the underlying distributions of V-QoE and GP-QoE have the same shape in 90% of network conditions studied.

Moreover, this also signifies that, for the QoE of the cloud gaming applications that are studied using a small sample size, it is difficult to evaluate V-QoE and GP-QoE independently. Even though the assessing criteria of both of these metrics are independent, they still have the same distribution for most of the network conditions. As far as we know, this is the first study to report a distribution comparison of cloud gaming applications. Prior studies presented differences between V-QoE and GP-QoE using MOS, but these differences are not meaningful until the underlying distributions were scrutinised. Our findings also highlight the limitations of MOS as a QoE metric. Using MOS alone shows significant differences between V-QoE and GP-QoE of all GuT, but, when the distributions were analysed, most of them have the same distributions. Therefore, MOS alone to evaluate QoE using ordinal scales like ACR scales is not adequate.

## 7. Conclusions

In this paper, we presented three different dimensions of quality that can be useful to evaluate the QoE of cloud gaming applications. We conducted subjective testing to measure the QoE of games under realistic network scenarios that include Delay-Only, PLR-Only and a Mixed scenario. We reported our findings as MOS and evaluated that GP-QoE relates to the O-QoE more closely than v-QoE for Delay-Only and Mixed scenarios. Additionally, we extended our analysis to underlying QoE distributions of these quality

dimensions and found that, for 91% of the scenarios, GP-QoE and V-QoE do not have statistically significant differences between them. As far we know, this is the first study to evaluate the distributions of different quality dimension to assess the QoE of cloud gaming. The results signify that MOS alone is not an adequate proxy of the user experience in small sample investigations, and other QoE metrics like QoE distributions need to be evaluated to draw valid conclusions.

We also found that different network conditions affect different types of games differently. We reported that the slow-paced games such as FIFA are less affected by the Delay-Only but significantly degrade for larger magnitudes of PLR. In addition, we found that games with high content complexity, faster pace and high level of graphics.

Performed worst in changing network conditions. This was demonstrated using GTA V which consistently performed worst out of most contenders in all network conditions. For medium paced games with medium levels of content complexity such as CSGO, a lot of variations were seen in their GP-QoE and O-QoE for different network conditions. These findings are significant for the service providers to tune the network parameters above the threshold level to avoid customer annoyance. These findings are also significant for the game developers and cloud gaming platforms to adapt their infrastructure to tackle changing/degrading network conditions. Although significant contributions are made, the results are limited to one game of each category and can be expanded by employing more games of each category to make broader conclusions. Moreover, in post COVID-19 times, more subjects can be employed to extend the existing analysis and to conduct the testing process.

In the future, we want to extend our analysis further to derive mathematical relationships between QoE distributions and QoS parameters for cloud gaming. In addition, further probing is required to determine the KPIs of cloud gaming and then correlate them to the subjective QoE, in order to formulate objective QoE metrics for cloud gaming.

**Author Contributions:** The project was proposed by M.G.M. An experimental testbed was designed by A.W. and N.A. A.W. carried out subjective testing. N.A. implemented the QoS in NetEM. A.W. did the data analysis and result evaluation. J.S. supervised the project along with M.G.M. All authors contributed towards writing and reviewing of the final manuscript. All authors have read and agreed to the published version of the manuscript.

**Funding:** No external funding was given for this research.

**Acknowledgments:** We are grateful to the School of Electronic Engineering and Computer Science at the Queen Mary University of London for their help and support in this project. We extend our thanks to Raul Mondragon and Karen Shoop for their contributions to the project. We would also like to thank Teragence for their collaboration and providing us with the active measurement data.

**Conflicts of Interest:** There is no conflict of interest declared by this research or authors.

**Sample Availability:** Samples of the compounds are available from the authors.

### Abbreviations

The following abbreviations are used in this manuscript:

| | |
|---|---|
| QoS | Quality of Service |
| PLR | Packet Loss Ratio |
| QoE | Quality of Experience |
| KPIs | Key Performance Indicators |
| MOS | Mean Opinion Score |
| GuT | Game under Test |
| fps | frames per second |

| ACR | Absolute Category Rating |
| KQIs | Key Quality Indicators |
| RTT | Round Trip Time |
| ITU-T | International Telecommunication Union- Telecommunication |
| FPS | First Person Shooting |
| GTA V | Grand Theft Auto V |
| CSGO | Counter Strike Global Offensive |
| V-QoE | Video Quality |
| GP-QoE | Game Playability Quality |
| O-QoE | Overall Quality |
| BUD | Better User Datagram protocol |
| P2W | Poor-to-Worse |
| G2B | Good-to-Better |
| WSR | Wilcoxon Signed Rank |

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
