# Peer review of "Subjective Quality Assessment for Cloud Gaming"

_2571-8800, doi:10.3390/j4030031_

Round 1

Reviewer 1 Report

The paper applies subjective testing to evaluate the effect of various network conditions on the overall user experience (the video and game-playability) in cloud gaming or Gaming on Demand services.  The study uses three different games with different genre, popularity and content complexity. The empirical study has been conducted systematically, and the results are interesting.  However, there are several issues with the paper.

First, there is no clear justification for the selection of the three different types of network scenarios.  Also, it is not clear what features (e.g., flow-level features and connection-level features) of the network conditions affect the user experience. How significant and important such features?

The introduction section should be improved to highlight the exact (specific) limitations in the existing works.

A brief introduction to subjective testing and its comparison to objective testing would be helpful for the reader.

There is also a considerable number of typos and other linguistics errors. The following are just a few examples.

  1. Due to the increased customer interest, big tech-giants such as Microsoft and 36 Google have introduced their on cloud gaming platforms.[1]. -  their what?  two dots.
  2. ubjective testing – missing “s”
  3. as used in previous works [4] & [9] -  use of “&” and there is no full stop mark
  4. Section 0. How to Use this Template – remove this

Author Response

We thank the reviewer for their time and efforts to read our work and provide us with their insightful comments. The comments were very helpful and contributed in the improvement of the manuscript. The inline response to the comments are as below:

  1. First, there is no clear justification for the selection of the three different types of network scenarios.  Also, it is not clear what features (e.g., flow-level features and connection-level features) of the network conditions affect the user experience. How significant and important such features?

    response: The justification for the selection of these three network scenarios is added in Section 4 (see line 173-179). they are justified for their existence in real IP networks. For the flow-level or connection level features, the paper addresses to both flow-level and connection-level features at important QoS parameters. The distinction between these features was not explicitly identified or addressed. 

  2. The introduction section should be improved to highlight the exact (specific) limitations in the existing works.

    response: The introduction is changed in accordance to this comment. the limitation of the existing works and the contribution of this paper is laid the second last paragraph of the introduction section. (see line 68 to line 75).

  3. A brief introduction to subjective testing and its comparison to objective testing would be helpful for the reader.

    response: We added a paragraph in the introduction section talking about the comparison of subjective and objective studies. see line 49 to 59. 
  4. There is also a considerable number of typos and other linguistics errors. The following are just a few examples.

    response: the paper is thoroughly reviewed. JS (one of the co-author) is a native English speaker and has reviewed the paper for grammatical and language issues. 

Reviewer 2 Report

Major comments:

  1. I recommend that the authors show in the manuscript the 7-point ACR scale used by the subjects in the subjective test to rate the Video-QoE, GP-QoE and O-QoE of the games studied.
  2. It would be very interesting if the importance of subjective quality assessment versus objective quality assessment in cloud computing were briefly discussed in the Introduction section of the manuscript.

Minor comments:

  1. There are some typographical errors throughout the manuscript, and some punctuation mistakes, which should be corrected. See, for instance:
    1. Line 151 of page 4.
    2. Line 163 of page 4.
    3. Line 328 of page 9.
  2. Section “0. How to Use this Template” must be removed from the manuscript.

Author Response

We thank the reviewer for their time and effort to review our manuscript. The comments were very helpful and insightful to uplift the quality of the manuscript. The response to the comments are as follow:

  1. I recommend that the authors show in the manuscript the 7-point ACR scale used by the subjects in the subjective test to rate the Video-QoE, GP-QoE and O-QoE of the games studied.

    response: The ACR is added as a figure, see figure 3 page 7. 

  2. It would be very interesting if the importance of subjective quality assessment versus objective quality assessment in cloud computing were briefly discussed in the Introduction section of the manuscript.

    response: A paragraph is added in the introduction to address a comparison between objective and subjective QoE evaluation. see line 49 to 59. 

  3. There are some typographical errors throughout the manuscript, and some punctuation mistakes, which should be corrected. See, for instance:

    response: the manuscript is thoroughly reviewed for grammatical and language issue. it was reviewed by Native English speaker to address any typographical errors.